# Endogenous Ethanol and Triglyceride Production by Gut *Pichia kudriavzevii*, *Candida albicans* and *Candida glabrata* Yeasts in Non-Alcoholic Steatohepatitis

**DOI:** 10.3390/cells11213390

**Published:** 2022-10-27

**Authors:** Babacar Mbaye, Patrick Borentain, Reham Magdy Wasfy, Maryam Tidjani Alou, Nicholas Armstrong, Giovanna Mottola, Line Meddeb, Stéphane Ranque, René Gérolami, Matthieu Million, Didier Raoult

**Affiliations:** 1IHU Méditerranée Infection, 19-21 Boulevard Jean Moulin, 13005 Marseille, France; 2Microbes Evolution Phylogeny and Infections (MEPHI), Institut de Recherche Pour le Développement, Aix-Marseille Université, 13005 Marseille, France; 3Unité Hépatologie, Hôpital de la Timone, APHM, 13005 Marseille, France; 4Assistance Publique-Hôpitaux de Marseille, 13005 Marseille, France; 5Laboratoire de Biochimie, Hôpital de la Timone, APHM, 13005 Marseille, France; 6C2VN, INSERM 1263, INRAE 1260, Team 5, Aix-Marseille Université, 13005 Marseille, France; 7VITROME: Vecteurs-Infections Tropicales et Méditerranéennes, Institut de Recherche Pour le Développement, Assistance Publique-Hôpitaux de Marseille, Service de Santé des Armées, Aix Marseille Université, 13385 Marseille, France

**Keywords:** nonalcoholic steatohepatitis, fructose, ethanol, *Pichia kudriavzevii*, *Candida*, yeast, fungi, auto-brewery syndrome, gut microbiota, gut mycobiome, metabolic syndrome, nonalcoholic fatty liver disease, metabolic-associated fatty liver disease, microbial culturomics

## Abstract

Nonalcoholic steatohepatitis (NASH) increases with fructose consumption and metabolic syndrome and has been recently linked with endogenous ethanol production, notably by high alcohol-producing *Klebsiella pneumoniae* (HiAlc Kpn). *Candida* yeasts are the main causes of auto-brewery syndromes but have been neglected in NASH. Here, the fecal ethanol and microbial content of 10 cases and 10 controls were compared. Ethanol was measured by gas chromatography-mass spectrometry. Species identification was performed by MALDI-TOF MS, and triglyceride production was assessed by a colorimetric enzymatic assay. The fecal ethanol concentration was four times higher in patients with NASH (median [interquartile range]: 0.13 [0.05–1.43] vs. 0.034 [0.008–0.57], *p* = 0.037). Yeasts were isolated from almost all cases but not from controls (9/10 vs. 0/10, *p* = 0.0001). *Pichia kudriavzevii* was the most frequent (four patients), while *Candida glabrata*, *Candida albicans,* and *Galactomyces geotrichum* were identified in two cases each. The concentration of ethanol produced by yeasts was 10 times higher than that produced by bacteria (median, 3.36 [0.49–5.60] vs. 0.32 [0.009–0.43], *p* = 0.0029). Using a 10% D-fructose restricted medium, we showed that NASH-associated yeasts transformed fructose in ethanol. Unexpectedly, yeasts isolated from NASH patients produced a substantial amount of triglycerides. *Pichia kudriavzevii* strains produced the maximal ethanol and triglyceride levels in vitro. Our preliminary human descriptive and in vitro experimental results suggest that yeasts have been neglected. In addition to *K. pneumoniae*, gut *Pichia* and *Candida* yeasts could be linked with NASH pathophysiology in a species- and strain-specific manner through fructose-dependent endogenous alcohol and triglyceride production.

## 1. Introduction

Carbohydrate-dependent liver steatosis was first described many years ago [1]. This led to the recent identification of fructose as a major mediator of nonalcoholic fatty liver disease, including nonalcoholic steatohepatitis (NASH) [2]. Carbohydrate-dependent steatosis has increased in parallel with the increase in fructose consumption, particularly in sweetened beverages and high fructose corn syrup (HFCS), which is correlated with the epidemic of obesity and type II diabetes [2]. Interestingly, carbohydrate-dependent steatosis is also accompanied by a significant increase in plasmatic triglycerides, whose role in liver steatosis has not been perfectly decrypted [2,3].

The other major metabolic liver disease is alcoholic liver disease, which also presents a succession of steatosis, fibrosis, cirrhosis, and eventually hepatocarcinoma [4]. Recently, the auto-brewery hypothesis has been revived by studies of the microbiota of patients with NASH [5,6]. Some studies have shown that NASH and auto-brewery syndromes (ABS) can occur after broad-spectrum antibiotic treatment (Appendix A). Reviewing 59 ABS cases from the literature, we found yeasts in most cases (*Candida* in 78%, *Saccharomyces* in 19%, and *Pichia* in 5%) and *Klebsiella pneumonia* in 3% of cases (just 2 cases and only a single case without any gut yeast) ([5], Appendix A).

At the same time, studies have shown a modification of the microbiota in NASH patients, particularly the identification of *K. pneumoniae*, whose in vitro analysis showed its capacity to produce alcohol in NASH patients [5]. This alcohol was also detected in the blood and stools of patients with NASH who, by definition, do not consume alcohol [5,7]. Nonsystematic works have also highlighted the possible association of yeasts with NASH and auto-brewery syndromes [5,8]. This is not surprising given that the first analysis of alcoholic fermentation reported by Lavoisier in 1789 was carried out with yeasts consuming sugar. Yeasts have been at the heart of human alcohol production for hundreds of years [9,10]. In addition to ethanol, triglycerides have been implicated in the pathophysiology of NASH [2,3]. However, in recent works linking yeast and NASH [8], the yeasts were not cultured, and ethanol and triglyceride production by gut yeast strains from NASH patients was not assessed.

In this work, we wanted to highlight the link that could exist between the microbiota, particularly fungi, and their production of alcohol and triglycerides based on fructose by using organisms specifically isolated from NASH patients.

## 2. Materials and Methods

### 2.1. Patients

We performed a case-control study comparing the fecal microbiota of 10 consecutive patients with nonalcoholic steatohepatitis (NASH) and 10 healthy controls. Patients were recruited from the Hepatology Department of Marseille University Hospital (Southeastern France). Healthy controls without liver disease were recruited from the same city using a snowball approach. Alcoholism, defined as drinking more than two alcoholic beverages per day for men and more than one alcoholic beverage per day for women, was an exclusion criterion for both cases and controls. Screening for diabetes, hypertension, elevated cholesterol, and hypertriglyceridemia was routine for cases. A history of diabetes, hypertension, elevated cholesterol, or hypertriglyceridemia were exclusion criteria for controls. In contrast, no blood pressure measurements or blood tests (to exclude diabetes or dyslipidemia) were performed in the controls. For patients, diagnostic criteria included (1) a metabolic syndrome (central obesity, increased triglycerides, reduced HDL cholesterol, high blood pressure, and type 2 diabetes), (2) steatohepatitis defined by the association of steatosis (liver steatosis at echography) with hepatitis (increased alanine aminotransferase (ALAT) more than two times the upper limit of normal and/or severe fibrosis), and (3) exclusion of differential diagnosis (notably alcoholic liver disease (>210 g/week in men or >140 g/week in women), chronic viral hepatitis (HBV, HCV), autoimmune hepatitis, hemochromatosis, and drug-induced liver disease). For cases and controls, exclusion criteria included antibiotics in the previous month, increased alcohol consumption (men ≥ 30 g/j, women ≥ 20 g/j), liver disease other than NASH, therapy that may cause steatosis (corticosteroid, amiodarone, estrogen, tamoxifen, and HIV protease inhibitors) and refusal or the impossibility of obtaining patient consent.

### 2.2. Measurement of Fecal Ethanol

One gram of stool was suspended in 5 mL of high-performance liquid chromatography (HPLC) water in HeadSpace glass vials. A calibration range was prepared from an ethanol stock solution in water with concentrations ranging from 0.5 to 100 mM. Standards and samples were spiked with isopropanol at 100 mM (internal standard). Measurements were performed using an HS-GC-MS system (Perkin Elmer, Villebon sur Yvette, France) combining an HS40 headspace injector, a Clarus 500 gas chromatograph, and an SQ8 S mass spectrometer. All vials were positioned onto the headspace sample tray after homogenization by shaking. They were heated one by one at 60 °C for 10 min to vaporize the alcohols that were automatically transferred to the gas chromatography system by overpressure (1 min, 25 psi) followed by a 1.8 s depressurization (needle/transfer line/GC inlet at 70/80/150 °C). Alcohols were introduced into a ZB-BAC2 chromatography column (30 m, 0.32 mm ID, 1.2 μm; Phenomenex, Le Pecq, France) maintained at 40 °C and separated using Helium as a carrier gas at 10 psi. Compounds were individually monitored by mass spectrometry with a selected ion recording (SIR) method: ethanol m/z 31 and isopropanol m/z 45. The MS inlet line and electron ionization source were set at 150 °C. All data were collected and processed using Turbomass 6.1 (Perkin Elmer) software. Internal calibration was calculated using the peak areas from the associated SIR chromatograms.

### 2.3. Selective Yeast Culture

Yeast culture was conducted by inoculating three fungal media by suspending 0.3 g of stool with 1 mL of 1X PBS for each sample. We used two commercial media, Chromoagar Candida agar (Becton Dickinson, Le Pont de Claix, France) and Sabouraud dextrose agar (Becton Dickinson, Le Pont de Claix, France). We also used another medium, the FastFung medium, specifically developed in our center suitable for the growth of fastidious fungi with a higher fungal colony count and lower culture contamination rate [11]. Serial dilutions were performed from 100 µL of the stock solution. For each tube, 50 µL was spread on each agar and incubated at 30 °C for 48 h before identification by matrix-assisted laser desorption ionization–time of flight mass spectrometry (MALDI-TOF MS) using a Microflex mass spectrometer (Bruker Daltonics, Leipzig, Germany), as previously reported [12].

### 2.4. Selective Culture of Enterobacteria and Resistance to Ethanol

Following the study of Yuan et al. [5] that highlighted the importance of specific strains of *Klebsiella pneumoniae* that produced an elevated level of ethanol in patients with NASH, we looked at the Gram-negative bacteria growing on MacConkey medium with or without ethanol in the medium. Thus, we chose a selective medium for nonfastidious Gram-negative bacteria: MacConkey agar liquid and solid medium. Two different approaches were used:

In the first approach, direct inoculation of diluted fecal samples was performed on MacConkey agar. Briefly, 0.3 g of stool was resuspended in 1 mL of 1X PBS after 10-fold serial dilution with 100 µL of this suspension, and 50 µL of each dilution was spread on MacConkey agar (Sigma-Aldrich, Saint Quentin Fallavier, France) and aerobically incubated at 37 °C for 24 h.

In the second approach, an enrichment step in a liquid medium with rumen and blood using blood culture bottles previously emptied of their contents was included. Briefly, enrichment was performed by adding 200 µL of the initial suspension (0.3 g of stools in 1 mL PBS 1X) of each sample inoculated into vials of aerobic blood cultures (bioMérieux, Durham, NC, USA) containing 20 mL of MacConkey broth (Dominique Dutscher, Brumath, France) enriched with 4 mL of defibrinated sheep blood and 4 mL of sterile rumen juice for 10 days. At Day 1, Day 3, Day 7, and Day 10, 500 µL of the bottle content was sampled followed by ten 10-fold serial dilutions and inoculation on MacConkey agar (Dominique Dutscher, Brumath, France). This second approach was also performed by adding 5% and 10% ethanol to the bottle to identify alcohol-resistant nonfastidious Gram-negative bacteria. All colonies were identified using MALDI-TOF mass spectrometry as previously reported [13].

### 2.5. Measurement of Ethanol Production by Microorganisms

Yeast ethanol production was measured by inoculating 1 mL of a yeast suspension at 1.5 × 10^4^ cfu/mL into 20 mL of liquid Sabouraud broth (Dominique Dutscher, France) and incubating for 24 h at 30 °C. After incubation, 1 mL of each culture was placed in a glass vial tube to measure the ethanol concentration by headspace GC/MS as described above. All enterobacteria identified in the NASH patients and controls were grown in MacConkey broth (Dominique Dutscher, France) to measure their ability to produce ethanol. For this purpose, 1 mL of a bacterial suspension of 1.5 × 10^4^ cfu/mL was inoculated into 20 mL of liquid MacConkey broth and incubated for 24 h at 37 °C.

### 2.6. Ethanol and Triglyceride Production Assay on 10% D-Fructose

Beyond ethanol production by endogenous gut yeasts from NASH patients, we also investigated triglyceride production for all yeast strains isolated from included individuals and for a standard *Candida albicans* strain from our laboratory used as a control. Thus, we inoculated yeast at a cell density of 5 McFarland in triplicate into 5 mL of 10% D-fructose (Sigma-Aldrich) sterile water solution. After three days of incubation at 30 °C, 1000 µL of the supernatant was collected and analyzed on an Atellica^®^ Solution Immunoassay and Clinical Chemistry Analyser (Siemens Healthineers, Saint-Denis, France) to measure triglycerides levels, following the manufacturer’s instructions.

### 2.7. Statistical Analysis

To compare quantitative variables, the bilateral unmatched Mann-Whitney test was performed with GraphPad Prism version 9 for Windows (GraphPad Software, San Diego, CA, USA). A *p*-value ≤ 0.05 was considered significant.

## 3. Results

### 3.1. Human Descriptive Results

#### 3.1.1. Characteristics of the Ten Patients with NASH

The mean patient age was 70.3 years (±7.6), and 4 out of 10 patients were male (Table 1). The mean body mass index was 27.6 (±3.2); seven patients had diabetes, eight had hypertension, and four had dyslipidemia. Two patients had F0/F1 fibrosis assessed by pulse elastometry fibroscan^®^ (ECHOSENS, Paris, France). The remaining eight patients had cirrhosis, of which one was assessed by pulse elastometry fibroscan^®^, three patients with compensated cirrhosis were assessed by clinical, biological, and morphological criteria, and four patients with decompensated cirrhosis were evaluated for liver transplantation. The characteristics of the patients and controls are described in Table 1.

#### 3.1.2. Increased Fecal Ethanol Concentration in NASH

The fecal ethanol concentration was four times higher in patients with NASH (median [interquartile range]: 0.13 [0.05–1.43] g/L) than in the 10 healthy controls (0.034 [0.008–0.57], two-tailed Mann-Whitney test, *p* = 0.037, Figure 1, Appendix A).

#### 3.1.3. High Prevalence of Gut Yeasts in NASH

In cultures on Chromoagar, FastFung, and Sabouraud media, we did not find any yeast in the 10 controls. However, at least one yeast was cultured in 9 of 10 NASH patients, and this was very significantly different from the controls (9/10 vs. 0/10, two-tailed Fisher test, *p* = 0.0001). One patient had both *Candida* (*C. glabrata*) and *Galactomyces geotrichum*. At the species level, *Pichia kudriavzevii* was the most frequently isolated yeast (four patients), while *Candida glabrata, Candida albicans,* and *Galactomyces geotrichum* were identified in two patients (Figure 2a, Appendix A).

#### 3.1.4. Increased Diversity of Potential Gram-Negative Pathogens in NASH

At the individual level, the number of bacteria growing in MacConkey broth enriched with blood and rumen (see methods) and agar with or without ethanol were not different between the two groups (median: two species per individual for both groups). Overall, the diversity of Gram-negative bacteria isolated was higher in patients with NASH. Indeed, 14 distinct species were isolated in the 10 patients with NASH compared with only 6 in the 10 controls (Figure 3, Appendix A).

### 3.2. In Vitro Experimental Models

#### 3.2.1. In Vitro Experimental Model to Assess Ethanol Production of Gut Yeasts

In an in vitro experimental model, ethanol production was evaluated on the 10 yeast strains isolated from NASH patients. This production was particularly notable for strains of *P. kudriavzevii*, *C. albicans,* and *C. glabrata* species capable of reaching levels of 6.3 g/L, 6.1 g/L, and 4.6 g/L, respectively. A large variation between strains of the same species was observed. Indeed, for the same species, ethanol production ranged from 1.4 to 6.3 g/L for *Pichia kudriavzevii*, 0.15 to 6.1 for *C. albicans,* and 3.5 to 4.6 for *C. glabrata*, suggesting a considerable strain-dependent effect for ethanol production. In contrast, *G. geotrichum* isolated from two patients produced negligible amounts of ethanol (Figure 2, Appendix A).

#### 3.2.2. In Vitro Experimental Model Assessing the Dependance and Resistance to Ethanol of Enterobacteria Isolated in NASH

There were more species resistant to 5% ethanol in NASH patients (Figure 3). Indeed, three species were found in both groups able to grow with 5% ethanol in the medium: *Enterobacter cloacae*, *Escherichia coli*, and *Klebsiella pneumoniae*. At the same time, two ethanol-resistant species were found only in patients with NASH: *Klebsiella oxytoca* and *Citrobacter sedlakii*. The latter species grew only in the presence of ethanol and was considered an ‘ethanophile’ strain. No strain was resistant to 10% ethanol. 

Several *Citrobacter* species sensitive to 5% ethanol were found in both NASH patients and controls. In addition, several ethanol-sensitive species were specific to NASH, such as *Hafnia alvei*, *Proteus mirabilis*, *Pseudomonas aeruginosa,* and *Raoultella planticola*. It is interesting to note that *Pseudomonas aeruginosa* and *Proteus mirabilis* are well-known human pathogens, while *Hafnia alvei* and *Raoultella planticola* are increasingly recognized as emerging human pathogens [14,15].

#### 3.2.3. In Vitro Experimental Model Assessing the Production of Ethanol Enterobacteria Isolated in NASH

In an in vitro experimental model, ethanol production was assessed for 36 available bacterial strains isolated on MacConkey medium corresponding to 11 Gram-negative species (Figure 4a, Appendix A). The six strains producing the highest levels of ethanol were all isolated from NASH patients. Surprisingly, two strains producing exceptional amounts of ethanol (Figure 4b, red arrow) were isolated from NASH patients and corresponded to the *Klebsiella pneumonia* species. At the species level, only four species produced more than 0.2 g/L: *K. pneumoniae*, *E. coli*, *Hafnia alvei,* and *Klebsiella oxytoca*. A visual examination of ethanol production according to the strains (Figure 4a) showed three strain populations, namely, low ethanol producers (<0.2 g/L), strains with high production (>0.2 g/L), and strains with exceptional ethanol production (>0.6 g/L).

We observed that strains from NASH patients produced more ethanol than strains from controls for *K. pneumoniae*, *K. oxytoca*, and *H. alvei* species (Appendix A). However, this was not the case for *E. coli,* for which no difference was observed. Finally, we compared ethanol production for species including at least three strains (Figure 4c). Only *Klebsiella pneumoniae* and *Escherichia coli* were associated with substantial ethanol production (mean between 0.3 and 0.5 g/L), while *Enterobacter cloacae* and *Citrobacter freundii* produced negligible amounts of ethanol. Collectively, these results suggest that among Gram-negative bacteria, *Klebsiella* is the genus most involved in endogenous ethanol production in NASH patients.

#### 3.2.4. Comparison of Ethanol Production between Gut Yeast and Enterobacteria In Vitro

Endogenous ethanol production by enterobacteria has also been evaluated extensively and characterized in NASH [5]. Here, we identified yeast only in NASH patients, while enterobacteria were detected in both NASH patients and controls. In an in vitro experimental model, we found high endogenous ethanol production by yeast at high concentrations depending on species and strains. To assess the relative role of yeast and enterobacteria in endogenous ethanol production in NASH, we compared the concentration of ethanol produced by yeast versus the concentration of ethanol produced by enterobacteria isolated from the 10 NASH patients included in the present study.

In total, 10 yeast strains producing a detectable amount of ethanol were isolated from nine NASH patients, and 21 strains of enterobacteria producing a detectable amount of ethanol were also identified from the NASH patients. The concentration of ethanol produced by yeast was 10 times higher than that produced by bacteria (median, [interquartile range]: 3.36 [0.49–5.60] for yeast versus 0.32 [0.009–0.43] for bacteria, Mann-Whitney test, *p* = 0.0029, Figure 5). For 7 of 10 gut yeasts, this production was much higher than that of the *K. pneumoniae* strain with the maximal ethanol production (1.09 g/L). This was strain dependent as one *C. albicans* strain produced a negligible amount of ethanol (Candida_albicans_N6, 0.15 g/L). In this patient, a high alcohol-producing *K. pneumoniae* strain (HiAlc Kpn) was identified (Klebsiella_pneumoniae_N6, 0.90 g/L, Figure 4a). In the one NASH patient who did not have yeast (Nash10), two enterobacteria were isolated: *Citrobacter youngae* and *Escherichia coli*. Unfortunately, the ethanol production of these two strains could not be evaluated.

Finally, Takahashi et al. [16] reported a synergistic effect of a strain of EtOH-producing enterobacteria, *Klebsiella pneumoniae,* and an EtOH-producing yeast, *Candida albicans*, isolated from an ABS patient. In this study, the coculture of *C. albicans* and *K. pneumonia* produced considerably more alcohol than *C. albicans* alone [16]. Strikingly, our patient with the highest fecal ethanol concentration (Nash7, fecal EtOH = 2.3 g/L) harbored a high ethanol-producing *Candida glabrata* (*C. glabrata*_N7, EtOH 4.6 g/L, Figure 2b) as well as the highest ethanol-producing bacterial strain, *Klebsiella pneumonia* (*K. pneumoniae*_N7, EtOH 1.09 g/L).

#### 3.2.5. In Vitro Experimental Model Assessing Triglyceride Production by Gut Yeasts

The species most involved in this triglyceride production were *Pichia kudriavzevii*, *C. glabrata,* and *C. albicans*, reaching levels of 3.9 mmol/L, 1.5 mmol/L, and 0.7 mmol/L, respectively. A large variation in triglyceride production between strains of the same species was observed (Figure 6). The production was highest for *Pichia kudriavzevii* species, ranging from 2.4 to 3.9 mmol/L, while *C. glabrata* produced 1.1 to 1.5 mmol/L, and *C. albicans* produced 0.6 to 0.8 mmol/L. In contrast, the isolated *Galactomyces geotrichum* strains produced low amounts of triglycerides (Figure 6, Appendix A). These results suggest that among the isolated yeasts, *Pichia kudriavzevii* appears to be the most involved in triglyceride production in NASH patients.

#### 3.2.6. In Vitro Experimental Model Assessing Fructose as a Specific Substrate for Ethanol Production

We then tested the functional link between fructose consumption, specifically associated with NASH in the literature [2], and ethanol production by an in vitro model. In this model, strains were grown on a medium containing only water and 10% D-fructose. Only nine strains were studied. Indeed, the Galactomyces_geotrichum_N9 strain was lost despite several culture attempts. We observed that 10% D-fructose and water were sufficient for *Candida* and *P. kudriavzevii* to produce ethanol levels higher than the best HiAlc *K. pneumoniae* (up to >2 g/L, Figure 5, Appendix A). In particular, all four strains of *P. kudriavzevii* produced more than 1 g/L of ethanol from this fructose-restricted medium. For the nine strains for which ethanol production could be assessed on both Sabouraud’s medium and 10% D-fructose medium, ethanol production was significantly higher on Sabouraud’s medium (mean ± SD, 3.49 ± 2.46 g/L vs. 1.16 ± 0.61 g/L, paired two-tailed *t*-test, *p* = 0.015, Figure 7).

## 4. Discussion

This work allowed us to ascertain that patients with NASH commonly carry yeast in the gut and that these organisms are significantly associated with the presence of alcohol in the stool. Alcohol has very rarely been measured in human stool [5]. In this work, we were able to show that in controls without NASH, ethanol was not detected, whereas ethanol is present in the stool of patients with NASH. This simple measure could be a crucial element in the diagnosis and follow-up of patients with NASH. Thus, the association between yeast-related ABS and NASH seems significant. It is likely that the association between yeast and alcohol production is a key factor in the cause of NASH cases. Yeasts consume sugars, in particular glucose and fructose [17], while producing alcohol as has been known for centuries [9,10]. Furthermore, the production of triglycerides associated with NASH may also result from the fermentation of fructose by yeast [18].

Our results regarding the presence of yeasts suggest the possibility of eradicating yeast in the gastrointestinal tract as a novel method for managing NASH, emphasizing the need to determine whether such treatment could allow the elimination of alcohol from the gut. The link between fructose, yeast, and endogenous alcohol production suggests a link between the global explosion of fructose consumption in sweetened beverages [19] and the increase in the incidence of NASH [2]. Confirming such an association would require further epidemiological studies focusing on the fructose–NASH ratio and the fructose–endogenous ethanol ratio. It is possible that increased antibiotic use also promotes yeast colonization, and this is consistent with the fact that auto-brewery syndromes have been observed following various antibiotic therapies [20,21,22,23,24,25].

Recent reports have demonstrated the role of ethanol-producing bacteria in NASH, particularly enterobacteria such as *K. pneumoniae* [5] and *Citrobacter* [26]. These results reinforce the validity of the data replicated in our study. However, yeasts, clearly identified as a cause of ABS (Appendix A), have been completely neglected in most recent studies of NASH [5,26], even though they are microorganisms with a higher capacity for producing ethanol, even on a 10% D-fructose restricted medium, than bacteria (Figure 5). In the literature, two cases combining ABS and NASH proven by liver biopsy have been reported [5,27]. In addition to the study of Yuan et al. mentioned above [5], we found one case with the identification of the *Candida parapsilosis* yeast [27]. Gut yeasts could have a more important role than *K. pneumoniae* in endogenous ethanol production as we found that most gut *Candida* and *Pichia* yeast produced much higher ethanol levels (up to 6-fold) than the *K. pneumoniae* strain with the maximal ethanol production.

The study we report here is strengthened by the diversity of microorganisms that we included, involving both yeasts and bacteria. This seems all the more relevant, as we found two patients in the literature with ABS harboring both yeast and *Klebsiella pneumoniae* [16,23]. In addition, a synergistic effect was previously reported between *Klebsiella pneumoniae* and *Candida albicans* in an ABS patient [16].

Microbiological culturing was critical to the assessment of strain-dependent ethanol production. While recent studies report preliminary findings linking *Candida* yeasts and NASH using serology and metagenomics [8], the present study allowed the identification of viable and active bacteria and yeasts from living patients being treated for NASH. Accurate and rapid culture and identification of fungal species were essential to characterize yeast species and antifungal susceptibility. Our center was the first to offer matrix-associated laser desorption ionization spectrometry-time-of-flight for routine microbial identification in the diagnostic laboratory [13]. It is also a reference diagnostic laboratory for fungi for Southeastern France. Based on 6192 clinical yeast isolates, we have implemented fungal diagnostics using a comprehensive and updated MALDI-TOF MS database [12]. We also developed an optimized medium (the ‘FastFung’ medium) for the culture of fastidious fungi [11]. This allows us to rapidly, accurately, and confidently culture and identify the intestinal yeast species found in the present study.

The link between NASH and endogenous ethanol and triglyceride production was not demonstrated in the present study. However, our study is a strong argument for further testing this hypothesis. To demonstrate this link, the Bradford-Hill criteria could be considered as follows. (1) Strength: We found a dramatic effect size here as 9 out of 10 of the cases had intestinal yeast compared to none of the controls. Moreover, ethanol levels produced by yeasts were much higher than those observed with *K. pneumoniae*, recently linked with NASH in humans [5]. (2) Consistency/reproducibility: An association with intestinal yeast has already been found in other studies from other centers in other countries [8]. However, new studies using high-throughput fungal culture methods similar to those used in the present work [11,12,13] should be conducted in other centers and clarify if yeast is present in all or specific NASH patient types. Clearly, these preliminary results need to be replicated in larger studies with multivariate analyses taking into account more potential confounding factors, and appropriate corrections for multiple comparisons. (3) Specificity: The fungal species and strains must be identified. Indeed, one of the intestinal yeast species (*Galactomyces geotrichum*) and one of the two *C. albicans* strains isolated in the present study did not produce ethanol. (4) Temporality: This criterion could be evaluated in future in vitro or in vivo experimental models. (5) Biological Gradient: Future studies should evaluate a quantitative association between gut yeast concentration and ethanol and fecal triglycerides in NASH patients. (6) Plausibility and consistency with current knowledge are already demonstrated for endogenous ethanol production. (7) Analogy is provided in the present work with the auto-brewing syndrome. Indeed, in this disease, gut yeasts cause (neurological) disease through endogenous ethanol production, and antifungal treatment cures the disease. The analogy is also supported by the recent study linking ethanol-producing *K. pneumoniae* and NASH [5]. (8) Finally, the last criterion, reversibility, will be the most important: can antimicrobial treatment, based on the isolation of an ethanol-producing fecal microbe, reverse endogenous ethanol production and cure NASH patients? In addition to the study by Yuan et al. [5], our study is a call to action for active collaborations between nutritionists, liver disease specialists, microbiology laboratory, and infectious disease specialists.

Our preliminary results on endogenous triglyceride production from fructose by gut yeasts in NASH are completely new and decipher new hypotheses for gut microbiome-associated metabolic diseases. Future studies should better characterize acyl chain lengths and lipid droplets potentially produced by gut yeasts, and their role in the host’s health and metabolic diseases. For instance, the mechanism of lipid droplet formation by the yeast *Saccharomyces cerevisiae* and its Sei1/Ldb16 Seipin complex has been only recently decrypted [28]. Maximal cholesterol assimilation has been evidenced for *Pichia kudriavzevii* yeast [29] but the possible impact on human lipid metabolism has not been investigated. Studies are needed to test if antifungal treatment could influence plasmatic triglyceride levels and liver fat.

Here, we present an original observational human study using multi-domain (bacterial and fungal) microbial culturomics [30] with in vitro experimental findings. Previous studies on the link between yeast and NASH used DNA sequencing and serological approaches [8]. The unparalleled advantage of culture (culturomics) over DNA or RNA sequencing (metagenomics) is to obtain strains. Isolation of patient strains in pure culture is the best way to characterize them metabolically (ethanol and triglyceride production), decipher the strain-specific effect [31], and investigate their susceptibility to antimicrobial treatment, but also to demonstrate their effect in vivo on liver fat regulation in future studies.

To confirm the putative instrumental role of gut yeasts for fatty liver disease and NASH through ethanol and triglyceride production, human studies remain of foremost importance. However, in vitro and animal models could help decipher the mechanism and are critical for causal inference, and to test reversibility. Here, we developed several in vitro models to better characterize ethanol resistance, dependence, and production by gut enterobacteria and yeasts. We isolated an ‘ethanophile’ strain suggesting that future ‘microbial culturomics’ studies on NASH should include culture media enriched in ethanol. In addition to fructose, several media using simple and/or complex carbohydrates will be helpful to decipher the substrate specificity for ethanol production. Indeed, this could have a major impact on the therapeutic diet for NASH patients. Lastly, a recent study showed that a hyperglycemia-stimulating diet induces liver steatosis in sheep [32]. Such animal models will be another opportunity to test and confirm the functional evidence of the putative link between NASH and endogenous ethanol and triglyceride production. Chiu et al. [33] recently reported an in vivo experimental study with fecal microbiota transplantation of control and NASH human feces to germ-free C57BL/6JNarl mice. Mice fecal content was analyzed by culturomics. Yeasts were reported in mice receiving NASH feces but not in those receiving CTL feces. The difference was significant. Strikingly, this result was not even discussed as the authors focused only on bacteria. This confirms that it is time to stop overlooking yeasts in NASH.

We have previously published a paper on the specific effect of *Lactobacillus* strain on weight regulation [31]. Again, in the present study, in the context of another nutritional and metabolic disease (NASH), we decipher a specific microbial effect of intestinal yeast on ethanol and triglyceride production in NASH patients (*P. kudriavzevii* > *C. glabrata = C. albicans* > *G. geotrichum*). However, even though the sample size is small, our results strongly suggest that the resolution at the species level is not sufficient. Indeed, among different strains of the same species, the ability to transform Sabouraud medium (40% glucose + peptone) or 10% D-fructose into ethanol is different. In patient ‘NASH_6’, we identified a *C. albicans* strain (Candida_albicans_N6) with a low capacity to transform D-fructose into ethanol. However, his *Klebsiella pneumoniae* strain (K_pneumoniae_N6) produced more ethanol (0.90 g/L) than his yeast (C_albicans_N6, 0.16 g/L on Sabouraud medium and 0.69 g/L on 10% D-fructose medium). This is the only yeast strain for which ethanol production was higher on D-fructose medium than on Sabouraud medium, but both were very low compared to high alcohol-producing yeasts. Experimental variability or strain specificity could explain this discrepancy. Nevertheless, the specific observation (*K. pneumoniae > C. albicans*) in this patient suggests that neither domain (yeast nor bacteria) can be overlooked in NASH.

Further studies are needed, but the strain-specific effect is critical for two reasons. First, it highlights the limitations of 16 s amplicon sequencing studies whose resolution is not capable of distinguishing strains of the same species. Indeed, species determination based on 250 bp (as is typically done in Illumina MiSeq instruments) is equivocal for polyphyletic taxonomic groups. Second, strain specificity is a definitive argument for culturomics to complement metagenomic studies. Indeed, only culturomic studies allow the domestication of live strains then available for full characterization, including for ethanol production.

The limitations of our study include a limited number of patients, the fact that the controls were younger than the cases, the use of only two culture media (Sabouraud and 10% D-fructose), and the incomplete analysis of metabolites produced beyond ethanol and triglycerides. However, our study opens a new field. Future studies should include more patients, better matched, and more comprehensively investigate the substrates and products of NASH-associated yeasts. However, fructose and ethanol seem to be the most relevant substrate and product, respectively, given the state of knowledge and recent literature on NASH [2,5,7,8].

We can speculate that in the future, culturomics focused on ethanol-producing microbes (yeast and bacteria) could be part of the diagnostic workup of any patient with NASH. Once ethanol-producing strains are identified, precision and personalized medicine will allow clinicians to treat patients with appropriate antimicrobial molecules based on in vitro susceptibility testing and/or probiotics that interfere with ethanol production. For example, it has been demonstrated in vitro that *Lactobacillus paracasei*, a probiotic marketed worldwide and widely consumed, is able to decrease ethanol production by *Candida kefyr* in a species-specific manner [34]. If it is confirmed that microbial endogenous ethanol and triglyceride productions play a critical role in the pathophysiology of NASH, future management of NASH patients will necessarily include evaluation of the fecal microbiota (fungal and bacterial) and strain-specific treatment. Inspired by the study of Yuan et al. [5], our study confirmed the putative role of HiAlc strains of *K. pneumoniae* but expanded the field to the HiAlc fungi strains to offer great hope to NASH patients.

In conclusion, this work links the presence of *Candida*-type yeasts, alcohol, and triglyceride production in the digestive tract of living NASH patients. It opens avenues for the therapeutic management and prevention of this syndrome, whose very rapid increase seems to be linked to changes in dietary behavior. In addition, our findings suggest the possibility of specific, inexpensive therapeutics targeting yeasts producing alcohol as a method for treating ABS and NASH.

## Figures and Tables

**Figure 1 cells-11-03390-f001:**
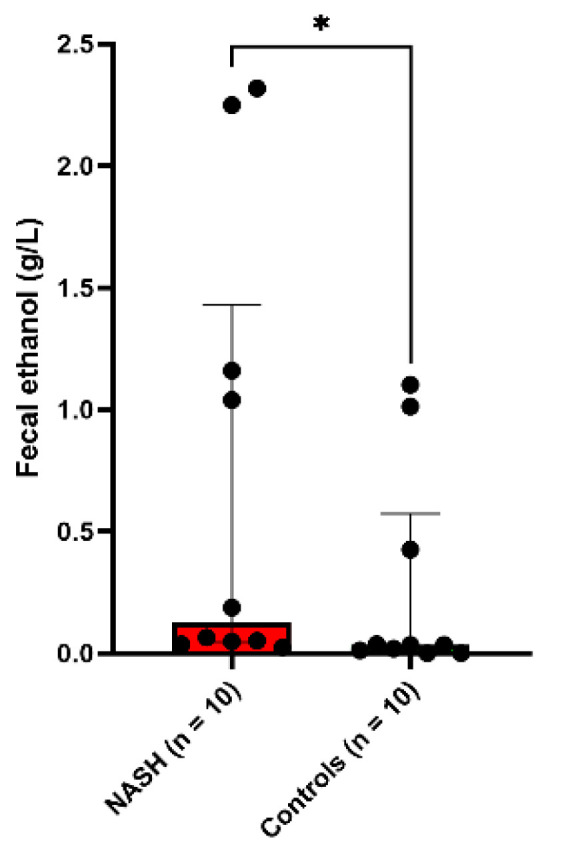
Increased fecal ethanol in NASH patients. * *p* < 0.05.

**Figure 2 cells-11-03390-f002:**
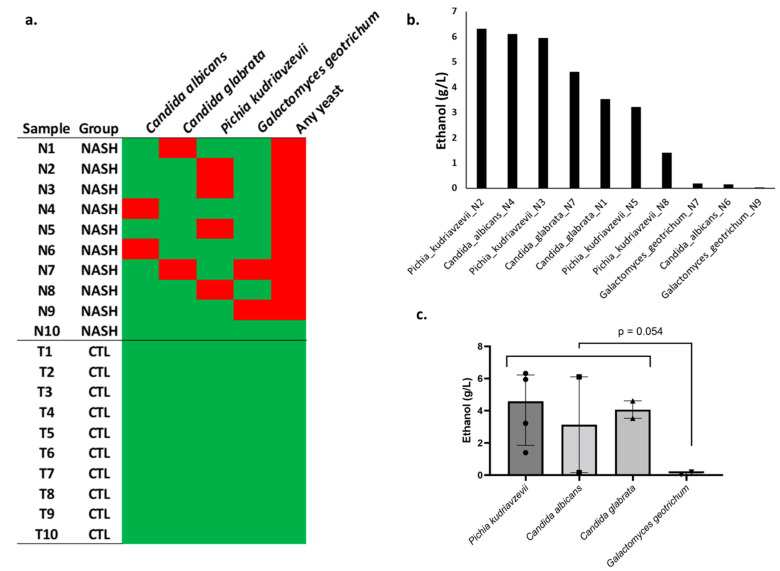
Isolation and ethanol production by yeast strains in an in vitro experimental model. (**a**) Yeast species isolated from each sample (N1, Patient Nash1, etc.). NASH: nonalcoholic steatohepatitis, CTL: healthy controls. Red square: isolation of yeast in culture. Green: no yeast isolated. (**b**) Ethanol production for each strain in an in vitro experimental model (Sabouraud medium, high-performance liquid chromatography); each strain was named with the species and the patient (for instance, Pichia_kudriavzevii_N2 is a strain isolated from patient Nash2). (**c**) Comparison between species.

**Figure 3 cells-11-03390-f003:**
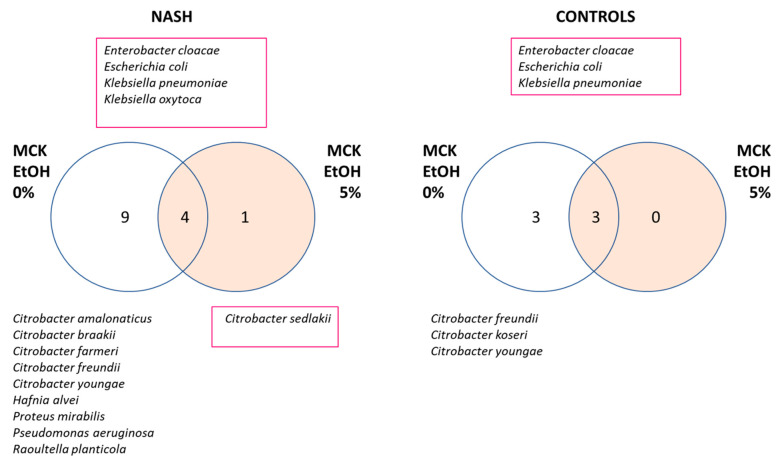
Increased diversity of Gram-negative bacteria in NASH patients. Strains were isolated using MacConkey medium. MCK: MacConkey medium, EtOH: ethanol, Species resistant to 5% ethanol were identified by a red square. A higher diversity of Gram-negative bacteria was observed in NASH patients.

**Figure 4 cells-11-03390-f004:**
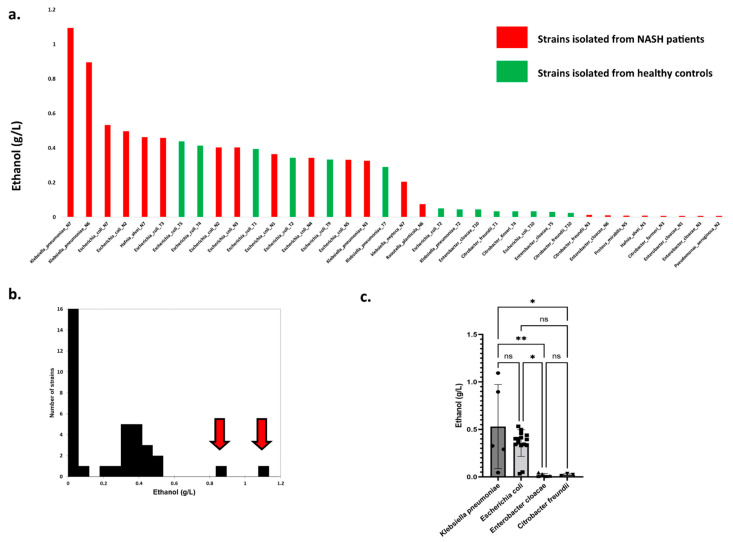
High ethanol-producing *Klebsiella pneumoniae* in NASH patients. (**a**) Production of ethanol by Gram-negative strains isolated in this study. The 2 strains producing the maximal ethanol concentration were both identified as *K. pneumoniae* from 2 different NASH patients (N6 & N7). (**b**) Distribution of strains according to ethanol production. Red arrows indicate two outliers with exceptional ethanol production. Both strains belong to the *K. pneumoniae* species and were isolated from NASH patients (N6 & N7). (**c**) Ethanol production according to bacterial species (species with at least 3 strains). * *p* < 0.05, ** *p* < 0.005.

**Figure 5 cells-11-03390-f005:**
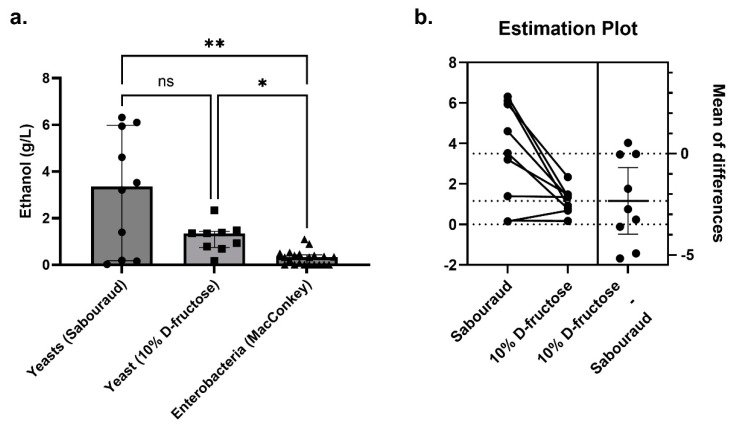
Ethanol production by yeast compared to enterobacteria isolated from NASH patients. (**a**) In Sabouraud’s medium (*n* = 10 yeast strains), the concentration of ethanol produced by yeast was 10-fold higher than that produced by enterobacteria (MacConkey’s medium, *n* = 21 bacterial strains), including *Klebsiella pneumoniae*, which produces a lot of alcohol (HiAlc Kpn, 1.09 g/L for the strain with maximum ethanol production). In a secondary analysis using 10% D-fructose restricted medium (*n* = 9 yeast strains, Galactomyces_geotrichum_N9 was lost before this secondary experiment), substantial ethanol production was observed. * *p* < 0.05, ** *p* < 0.005. (**b**) For the nine strains for which ethanol production could be assessed on both Sabouraud’s and 10% D-fructose, ethanol production was significantly higher on Sabouraud’s medium (paired two-tailed *t*-test, *p* = 0.015).

**Figure 6 cells-11-03390-f006:**
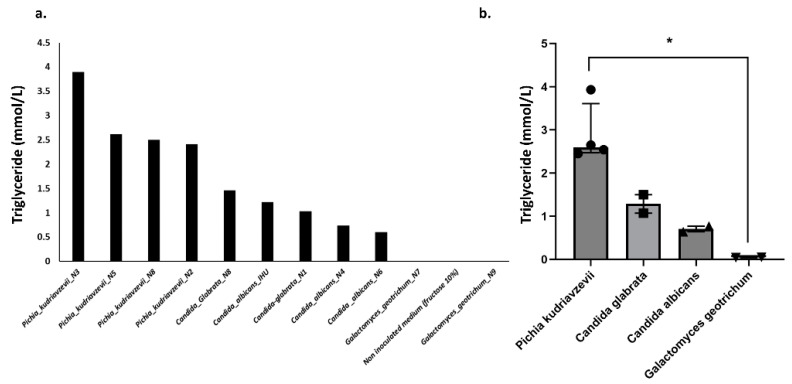
Triglyceride production by yeast strains. A value of 0.036 mmol/L was measured for ultrapure water and considered the background value. The values provided in the figure took into account this background value. (**a**) Triglyceride production for each strain, (**b**) Comparison between species. * *p* < 0.05.

**Figure 7 cells-11-03390-f007:**
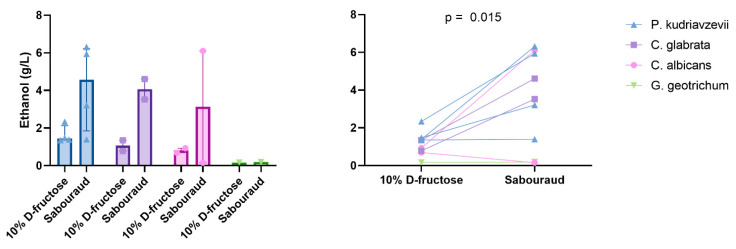
Ethanol production by gut yeasts according to species and medium. Median and interquartile are shown. Paired two-tailed *t*-test.

**Table 1 cells-11-03390-t001:** Patients’ and controls’ characteristics.

Patient	Sex	Age (Years)	Alcoholism ^a^	Diabetes	Hypertension	Increased Cholesterol ^b^	Hypertriglyceridemia ^c^	Weight ^d^
Nash1	M	58	No	Yes	Yes	No	No	Obesity
Nash2	M	69	No	Yes	Yes	Yes	No	Overweight
Nash3	F	66	No	Yes	Yes	Yes	No	Obesity
Nash4	M	83	No	No	Yes	No	No	Obesity
Nash5	F	71	No	Yes	Yes	Yes	Yes	Obesity
Nash6	F	92	No	Yes	Yes	No	No	Overweight
Nash7	F	64	No	Yes	Yes	Yes	Yes	Lean
Nash8	F	75	No	Yes	Yes	Yes	No	Overweight
Nash9	F	66	No	No	No	No	No	Lean
Nash10	M	62	No	Yes	Yes	Yes	Yes	Obesity
Ctrl1	M	24	No	No ^e^	No ^e^	No ^e^	No ^e^	Lean
Ctrl2	M	26	No	No ^e^	No ^e^	No ^e^	No ^e^	Lean
Ctrl3	M	38	No	No ^e^	No ^e^	No ^e^	No ^e^	Lean
Ctrl4	F	30	No	No ^e^	No ^e^	No ^e^	No ^e^	Lean
Ctrl5	M	27	No	No ^e^	No ^e^	No ^e^	No ^e^	Lean
Ctrl6	M	31	No	No ^e^	No ^e^	No ^e^	No ^e^	Lean
Ctrl7	F	44	No	No ^e^	No ^e^	No ^e^	No ^e^	Lean
Ctrl8	F	27	No	No ^e^	No ^e^	No ^e^	No ^e^	Lean
Ctrl9	F	24	No	No ^e^	No ^e^	No ^e^	No ^e^	Lean
Ctrl10	M	39	No	No ^e^	No ^e^	No ^e^	No ^e^	Lean

^a^ Defined as an individual who drank more than two alcoholic beverages a day for men and more than one alcoholic beverage a day for women M: Male, F: Female, ^b^ Hypercholesterolemia was defined by total cholesterol > 2 g/L or LDL-cholesterol > 1.6 g/L, ^c^ Hypertriglyceridemia was defined by triglycerides > 1.5 g/L. ^d^ Obesity: body mass index (BMI) ≥ 30 kg/m^2^, Overweight: BMI between 25 and 30 kg/m^2^, Lean: BMI ≤ 25 kg/m^2^. ^e^ No known medical history.

## Data Availability

Data supporting reported results can be found in the Appendix A.

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
