# Peer review of "Endogenous Ethanol and Triglyceride Production by Gut Pichia kudriavzevii, Candida albicans and Candida glabrata Yeasts in Non-Alcoholic Steatohepatitis"

_cells, 2022, doi:10.3390/cells11213390_

Round 1
Reviewer 1 Report
Comments to the author:
There appears to be an error in section 3.6 Triglyceride production by gut yeasts: the word “ethanol” was used, and I believe the word should be “triglyceride”
Also, the characteristics of the control group should be provided to show they are similar except for the NASH diagnosis. Did you ask subjects if they drank alcohol? This would be an important parameter as well.
Reviewer comments
The main question of the research is: what is the link between the gut microbiome and non-alcoholic fatty liver disease (NAFLD)? The researchers specifically looked at patients with nonalcoholic steatohepatitis (NASH) and the relationship to alcohol- and triglyceride-producing microbes, specifically yeast.
This is a particularly important topic and quite original because the researchers use a group of patients with NASH compared to a control group. Hypotheses have been advanced that NAFLD may be a function of endogenous ethanol production, but very few studies have been conducted that test the hypothesis in human subjects.
The research appears to conform to international standards for inclusion of human subjects and there appear to be no conflict of interests. The paper is well-written with clarity and depth, and multiple appropriate references. The authors describe their case-control method that includes 10 subjects with NASH and 10 healthy subjects in a control group. They provide detail with references for the materials and methods including the measurement of fecal alcohol and the culture methods for yeast and Enterobacteria as well as the statistical methods used. The tables and figures add to the clear delineation of the results and assist the reader in deciphering the data.
Results: First, the authors detail an analysis of the characteristics of the NASH group. It would be important to know the characteristics of the control group to see if the groups are similar except for NASH. It would also be important to know if any of the subjects drink alcohol on a regular basis.
On examination of the stool cultures, significant differences were discovered between the NASH group and the control group. Data demonstrate a four-times higher concentration of fecal alcohol in the NASH group with 9 of the 10 subjects having at least one yeast, but none of the control group having yeast. The NASH group had cultures of Candida glabrata, Candida albicans, and Galactomyces geotrichum but Pichia Kudriavzevii was the most frequently found yeast species.
Furthermore, the diversity of Gram-negative bacteria was higher in the NASH group, with 14 distinct species isolated in the group of 10 NASH subjects and 6 isolated in the control group of 10 subjects. Interestingly, enterobacteria were detected in both groups but the amount of ethanol produced by yeast (10 strains found only in the NASH group) was 10 times higher than that produced by enterobacteria. Yeast strains from the NASH group also showed higher triglyceride production with the highest rate by Pichia.
The discussion section asserts the significance of a “crucial element” for diagnosis of auto-brewery syndrome (ABS) and NASH by measuring alcohol in the stool. Alcohol is present in the stool of the patients with NASH but not in the control group. The authors add discussion about the alcohol-producing enterobacteria and triglyceride production in NASH patients. Triglycerides may well be produced by yeast. The links demonstrated so far between diet, yeast, bacteria, ABS, and NASH provide possibilities for inexpensive treatments for the disorders.
These results help answer the main question of the link between the microbiomes and ethanol and triglyceride production in persons with NASH compared to a control group. This research is a small beginning that contributes to our understanding of the mechanism of NASH. It will also serve as an impetus to future large-scale studies to confirm or refute the findings. Studies like this will enlarge our understanding of ABS as well as the growing problem of liver disease in non-drinkers.
Reviewer 2 Report
In the current manuscript by Mbaye et al., the authors investigated the possibility that ethanol-producing gut bacteria may contribute to the development and/or pathology of non-alcoholic steatohepatitis (NASH). Many studies have shown that NASH patients demonstrate gut dysbiosis, which contributes to the development of NASH mainly by affecting the host immunity. The authors identified several ethanol- or triglyceride (TG)-producing gut bacteria in the NASH gut microbiota and suggested a link between the NASH pathophysiology and these ethanol- or triglyceride-producing gut bacteria. This reviewer does not think that the authors have successfully provided sufficient evidence in order to demonstrate the link between NASH and endogenous ethanol and triglycerides production, however, the hypothesis is quite interesting and would be worth publishing at this stage.
Major
1. As for the statistical analysis, please perform multiple comparison with appropriate corrections in order to avoid making errors by repeating comparisons between two groups.
2. As for the TG analysis, in the NASH liver, TGs with long acyl chains are accumulated as lipid droplets. Thus, the current analysis is not convincing because the authors have not provided information of acyl chain lengths that were produced by the bacteria. Can the authors provide information about the acyl chain lengths? Or, at least provide discussion or future plans?
3. It seems that TG-producing bacteria contribute to the development of steatosis rather than hepatitis, because TGs accumulate as lipid droplets in hepatocytes, right? Adding discussion about to which pathologies (i.e., steatosis and hepatitis) TG-producing bacteria and ethanol-producing bacteria could contribute would strengthen the manuscript.
Minor
1. In the introduction, does “Glucido” mean “glucide” or “glucose”? It does not seem to be an English word.
2. Figure legend of Fig. 5 is missing.
Reviewer 3 Report
The underlying etiology of nonalcoholic steatohepatitis (NASH) is believed to be quite varied. Excessive production of endogenous alcohol has been recently shown to play a pathological role. This work links the presence of Candida-type yeasts with production of alcohol and tri-glyceride in the digestive tract of NASH patients. This study possibly reveals the contribution of yeast to NASH pathogenesis. However, this study is limited by the descriptive feature, and verification of functional role of yeast in NASH needs to be provided.
Major concerns:
1. NASH patients in this paper were selected from specific population. Is yeast present in all or most NASH patient types?
2. Yeasts were isolated from the NASH patients, were these yeasts previously identified?
3. Did the authors compare the capacity of Klebsiella pneumoniae (HiAlc Kpn) producing ethanol with that of yeast?
4. In vivo functional evidence demonstrating the effect of yeast on ethanol and TG production is lacking.
Round 2
Reviewer 3 Report
I expect this work to be well demonstrated in vivo data in future.
